# First Detection and Genetic Identification of *Wolbachia* Endosymbiont in Field-Caught *Aedes aegypti* (Diptera: Culicidae) Mosquitoes Collected from Southern Taiwan

**DOI:** 10.3390/microorganisms11081911

**Published:** 2023-07-27

**Authors:** Li-Lian Chao, Chien-Ming Shih

**Affiliations:** 1Program in Tropical Medicine, College of Medicine, Kaohsiung Medical University, Kaohsiung 807, Taiwan; d91632003@gmail.com; 2Graduate Institute of Medicine, Kaohsiung Medical University, Kaohsiung 807, Taiwan; 3Department of Medical Research, Kaohsiung Medical University Hospital, Kaohsiung 807, Taiwan

**Keywords:** *Wolbachia*, *wsp* gene, *Aedes aegypti*, mosquito, Taiwan

## Abstract

The prevalence and genetic character of *Wolbachia* endosymbionts in field-collected *Aedes aegypti* mosquitoes were examined for the first time in Taiwan. A total of 665 *Ae. aegypti* were screened for *Wolbachia* infection using a PCR assay targeting the *Wolbachia* surface protein (*wsp*) gene. In general, the prevalence of *Wolbachia* infection was detected in 3.3% *Ae. aegypti* specimens (2.0% female and 5.2% male). Group-specific *Wolbachia* infection was detected with an infection rate of 1.8%, 0.8% and 0.8% in groups A, B and A&B, respectively. Genetic analysis demonstrated that all *Wolbachia* strains from Taiwan were phylogenetically affiliated with *Wolbachia* belonging to the supergroups A and B, with high sequence similarities of 99.4–100% and 99.2–100%, respectively. Phylogenetic relationships can be easily distinguished by maximum likelihood (ML) analysis and were congruent with the unweighted pair group with the arithmetic mean (UPGMA) method. The intra- and inter-group analysis of genetic distance (GD) values revealed a lower level within the Taiwan strains (GD < 0.006 for group A and GD < 0.008 for group B) and a higher level (GD > 0.498 for group A and GD > 0.286 for group B) as compared with other *Wolbachia* strains. Our results describe the first detection and molecular identification of *Wolbachia* endosymbiont in field-caught *Ae. aegypti* mosquitoes collected from Taiwan, and showed a low *Wolbachia* infection rate belonging to supergroups A and B in *Ae. aegypti* mosquitoes.

## 1. Introduction

*Wolbachia* is a facultative intracellular and naturally occurring endosymbiont found in a wide range of terrestrial arthropods and nematodes [1,2,3,4,5]. This bacterium was first discovered in the reproductive tissues of the *Culex pipiens* mosquito, and *Wolbachia pipientis* was firstly described [6]. In the insect host, it is estimated to be naturally present in 60–76% of known species [7,8,9]. *Wolbachia* endosymbiont is not known to directly infect vertebrates and contains a powerful ability to manipulate the reproductive system in diverse ways, such as parthenogenesis, feminization of males and inducing cytoplasmic incompatibility (CI), which cause deleterious alterations of the reproductive system in invertebrate hosts that will lead to the suppression of vector populations and interference in pathogen transmission [10]. This fascinating aspect of its ability has inspired researchers targeting this endosymbiont for vector control. Indeed, this reducing ability of *Wolbachia* has been utilized to eradicate the mosquito species of *Culex pipiens fatigans* [11]. Most recently, *Wolbachia* strains of wMel and wAlbB have been successfully transfected into *Aedes aegypti* mosquitoes and shown to inhibit/reduce infections with zika, dengue, chikungunya, yellow fever and *Plasmodium* [12,13,14,15,16,17,18]. Although *Wolbachia* have demonstrated the detrimental role of blocking the transmission of mosquito-borne viruses [18,19,20,21,22,23], the existence and genetic identity of *Wolbachia* endosymbiont in field-caught *Aedes aegypti* mosquitoes of Taiwan has never been investigated.

The *Aedes aegypti* is a major mosquito species around the world and is incriminated as the transmission vector for several mosquito-borne viruses that infect humans, especially dengue, zika, yellow fever, and chikungunya viruses [23,24,25]. Although previous studies have claimed that the *Ae. aegypti* is not naturally infected with *Wolbachia* [7,26,27], recent investigations have provided solid evidence of natural *Wolbachia* infection detected in *Ae. aegypti*, including the detection of *Wolbachia* endosymbiont in the larvae and adults of field-collected *Ae. aegypti* from Malaysia, India, USA, and Philippines [28,29,30,31]. Thus, this evidence clearly demonstrates that the natural infection of *Wolbachia* endosymbiont in *Ae. aegypti* mosquitoes appears to be more commonly observed than previously described. However, there has been no research focusing on the genetic composition and affiliation of *Wolbachia* endosymbiont in field-caught *Ae. aegypti* mosquitoes in Taiwan.

The molecular approach provides the feasibility to differentiate the genetic variance at the individual base-pair level and provides a much more powerful method for discriminating the genetic diversity between and within supergroups of *Wolbachia* endosymbionts [32,33,34,35,36,37]. Current investigations focused on the molecular markers of *Wolbachia* surface protein (*wsp*) and 16S rDNA genes have demonstrated the existence of at least 16 supergroups [8,38,39,40,41]. The supergroups A and B are mainly found in arthropods, and may alter reproduction [42]. Thus, molecular analysis based on the genetic variation of the *wsp* gene has made it possible to facilitate the genetic discrimination of taxonomically similar *Wolbachia* endosymbionts within various mosquitoes.

It is postulated that the *Wolbachia* endosymbionts in field-caught *Ae. aegypti* mosquitoes of Taiwan may be genetically different from the existing common groups of *Wolbachia* throughout the world. Thus, the objectives of this study are to examine the presence of *Wolbachia* in field-caught *Ae. aegypti* from Taiwan and to determine the genetic identity of *Wolbachia* endosymbionts detected in *Ae. aegypti* mosquitoes. In addition, the phylogenetic affiliation of *Wolbachia* strains detected in *Ae. aegypti* mosquitoes of Taiwan was further analyzed by comparing their differential nucleotide composition with other *Wolbachia* strains described from various biological and geographical sources that have been documented in GenBank.

## 2. Materials and Methods

### 2.1. Field Collection and Genetic Identification of Mosquito Specimens

All specimens of adult *Ae. aegypti* mosquitoes investigated in this study were collected from three districts (Fongshan, Cianjhen, and Lingya) of Kaohsiung City, located in southern Taiwan (Figure 1). All these mosquitoes were captured from several places close to human houses by using ovitrap and photocatalyst mosquito traps for a period of four weeks. These traps were placed in or outside the house and were operated continuously overnight, from 16:00 p.m. to 08:00 a.m. the following morning. All mosquito specimens were identified to the species level based on their morphological characteristics, as previously described [43], and collected specimens were stored at −80 °C for further molecular analysis. The genetic identification of the mosquito species of southern Taiwan was compared with the sequences documented in GenBank and performed by targeting the mitochondrial CO1 gene.

### 2.2. DNA Extraction from Mosquito Specimens

Genomic DNA was extracted from individual mosquito specimens collected in this investigation. In general, the individual mosquito specimen was placed in a 1.5 mL microcentrifuge tube that was filled with 180 μL of lysing buffer solution equipped with a DNeasy Blood & Tissue Kit (catalogue no. 69506, Qiagen, Taipei, Taiwan), and then the samples were homogenized with a tissue homogenizer (TissueLyser II, Qiagen, Hilden, Germany), as instructed by the manufacturer. The homogenated fluid was further centrifuged at room temperature and the supernatant fluid was further processed by a DNeasy Blood & Tissue Kit, as instructed by the manufacturer. After filtration, the filtrated solution was collected with a second vial and the DNA concentration in the filtrated solution was measured with a microplate spectrophotometer (Epoch, Biotek, Shoreline, WA, USA), and the extracted DNA was stored at −80 °C for further analysis.

### 2.3. Wolbachia DNA Amplification via Nested Polymerase Chain Reaction (nPCR)

Extracted DNA samples from the mosquito specimens were used as a template for PCR amplification. Initially, the primer set of 81F (5′-TGGTCCAATAAGTGATGAAGAAAC-3′) and 691R (5′-AAAAATTAAACGCTACTCCA-3′) was used to amplify the universal *wsp* gene. A nested PCR was then performed using the group-A-specific primer set of 328F (5′-CCAGCAGATACTATTGCG-3′) and 691R, which amplified with a product of approximately 360 bp, and the primer set of 81F and 522R (5′-ACCAGCTTTTGCTTGATA-3′) served as the group-B-specific primers, which amplified with a product of approximately 440 bp, as previously described [37]. All PCR reagents and Taq polymerase enzymes were obtained and used as instructed by the supplier (Takara Shuzo Co., Ltd., Kyoto, Japan). The PCR amplification was performed with a thermocycler (Veriti, Applied Bioosystems, Taipei, Taiwan), and each 25 μL reaction mixture contained a 3 μL DNA template, 1.5 μL forward and reverse primers, 2.5 μL 10× PCR buffer (Mg^2+^), 2 μL dNTP mixture (10 mM each), 1 unit of Taq DNA polymerase and was filled up with an adequate volume of ddH_2_O. For comparison, adequate amounts of sterile distilled water were added in the reaction mixture for serving as a negative control. The PCR conditions were started with a pre-cycle of denaturation at 94 °C for 5 min and then amplified for 35 cycles with the conditions of denaturation at 94 °C for 1 min, annealing at 53 °C/55 °C for group A/B for 1 min, extension at 72 °C for 1 min, and a final extension step at 72 °C for 10 min. For visualizing the DNA products, all amplified products were electrophoresed on 1.5% agarose gels in Tris-Borate-EDTA (TBE) buffer and then the gels were stained with ethidium bromide. A 100 bp DNA ladder (GeneRuler, Thermo Scientific & Invitrogen, Taichung, Taiwan) was used as the standard marker for comparison. A negative control of distilled water was included in parallel with each amplification.

### 2.4. Genetic Identification of Mosquito Species

DNA samples extracted from *Wolbachia*-infected and uninfected mosquito specimens were used for identifying the genetic identity of tested mosquito by targeting the mitochondrial CO1 gene. The primer sets of CO1-F1/CO1-R1 were used to amplify the CO1 gene of mosquitoes, as described previously [44]. The PCR conditions for performing CO1 gene amplification were started with a pre-cycle of denaturation at 95 °C for 5 min and 5 cycles with the conditions of 94 °C for 40 s, 45 °C for 1 min, and 72 °C for 1 min. Thereafter, 35 cycles took place with the conditions of denaturation at 94 °C for 40 s, annealing at 52 °C for 1 min, extension at 72 °C for 1 min, and then a final extension step at 72 °C for 10 min. PCR amplification was performed with a thermocycler (Veriti, Applied Bioosystems, Taipei, Taiwan), and each 25 μL reaction mixture contained 3 μL DNA template, 1.5 μL forward and reverse primers, 2.5 μL 10× PCR buffer (Mg^2+^), 2 μL dNTP mixture (10 mM each), and 1 unit of Taq DNA polymerase and was filled up with an adequate volume of ddH_2_O. A negative control of distilled water was included in parallel with each amplification.

### 2.5. Phylogenetic Analysis Based on Wolbachia wsp Gene

Selective samples with clear bands on agarose gel were used for gene sequencing. In principle, 10 μL of each selective sample was submitted for DNA sequencing (Mission Biotech Co., Ltd., Taipei, Taiwan). After purification, sequencing reaction was performed with 25 cycles under the same conditions and the same primer set of the initial amplification of mosquito’s DNA with the dye-deoxy terminator reaction method using the Big Dye Terminator Cycle Sequencing Kit in an ABI Prism 377-96 DNA Sequencer (Applied Biosystems, Foster City, CA, USA). The determined sequences were initially edited with BioEdit software (V5.3) and aligned with the CLUSTAL W software [45]. Thereafter, the aligned sequences of *Wolbachia wsp* gene from Taiwan specimens were analyzed by comparing with other *Wolbachia* sequences containing 5 group A, 5 group B and 2 outgroup strains identified from the different biological and geographical origins documented in GenBank (Table 1). Phylogenetic analysis was performed with maximum likelihood (ML) compared with the unweighted pair group with arithmetic mean (UPGMA) method to estimate the phylogeny of the entire alignment using the MEGA X software package [46]. The genetic distance values of inter- and intra-species variations were also analyzed with the Kimura two-parameter model [47]. All phylogenetic trees were constructed and performed with 1000 bootstrap replications to evaluate the reliability of the construction, as described previously [48].

### 2.6. Nucleotide Sequence Accession Numbers

The nucleotide sequences of PCR-amplified *wsp* genes of 12 group A and 5 group B *Wolbachia* detected in field-caught *Ae. aegypti* mosquitoes of Taiwan have been registered and assigned the following GenBank accession numbers: group A of KH-FS-Ae-10409-F1 (OP882272), KH-FS-Ae-10410-F4 (OP882273), KH-FS-Ae-10410-F5 (OP882274), KH-FS-Ae-10410-F7 (OP882275), KH-FS-Ae-10410-F8 (OP882276), KH-FS-Ae-10410-F12 (OP882277), KH-FS-Ae-10411-M5 (OP882278), KH-FS-Ae-10411-M9 (OP882279), KH-FS-Ae-10411-M10 (OP882280), KH-FS-Ae-10411-M11 (OP882281), KH-FS-Ae-10411-M12 (OP882282), and KH-FS-Ae-10411-M13 (OP882283); group B of KH-FS-Ae-10410-F4 (OP896740), KH-FS-Ae-10411-M9 (OP896741), KH-FS-Ae-10411-M10 (OP896742), KH-FS-Ae-10411-M11 (OP896743) and KH-FS-Ae-10411-M12 (OP896744), respectively.

## 3. Results

### 3.1. Detection of Wolbachia in Field-Caught Ae. aegypti Mosquitoes

The presence of *Wolbachia* endosymbiont was detected in *Ae. aegypti* mosquitoes with a nested PCR assay targeting the group-specific *wsp* gene. The amplified products were visualized on gels with a molecular size of approximately 360 bp and 440 bp for group A and group B *Wolbachia*, respectively. The *Wolbachia* infection was detected in 3.3% (22/665) individual *Ae. aegypti* mosquitoes collected from Kaohsiung, Taiwan. An infection rate of 5.2% and 2.0% was detected in males and females, respectively (Table 2). In addition, group-specific *Wolbachia* infection with groups A, B and A&B was detected in 1.8% (12/665), 0.8% (5/665) and 0.8% (5/665) of mosquito specimens, respectively (Table 2).

### 3.2. Genetic Analysis of Wolbachia Detected in Field-Caught Ae. aegypti Mosquitoes

To identify the genetic identity of *Wolbachia* endosymbionts in *Ae. aegypti* mosquitoes of Taiwan, the *wsp* gene sequences of 12 group A and 5 group B Taiwan *Wolbachia* strains were aligned and analyzed with the downloaded *Wolbachia* sequences of 5 group A, 5 group B, and 2 outgroup strains from various origins documented in GenBank. The results revealed that all *Wolbachia* strains detected in Taiwan *Ae. aegypti* were genetically affiliated with the *Wolbachia* type strains of supergroups A (GenBank no. KY817476) and B (GenBank no. AF020059), with a high sequence similarity of 99.4–100% and 99.2–100%, respectively (Table 3 and Table 4). Based on the genetic distance (GD) values, the intra- and inter-species analysis revealed a lower level (GD < 0.006) of genetic divergence within the group A Taiwan strains as compared with the group B (GD > 0.498) and outgroup (GD > 0.788) *Wolbachia* strains (Table 3). In addition, a lower level (GD < 0.008) was observed within the group B Taiwan strains as compared with the group A (GD > 0.302) and outgroup (GD > 0.515) *Wolbachia* strains (Table 4).

### 3.3. Phylogenetic Analysis of Wolbachia Detected in Field-Caught Ae. aegypti Mosquitoes

Based on the sequence alignment of *wsp* genes, phylogenetic relationships were analyzed to reveal the genetic affiliation among 29 *Wolbachia* strains used in this study. The repeatability of the clustering specimens presented in phylogenetic trees was analyzed using bootstrap analysis. The phylogenetic relationships of group A and group B *Wolbachia* strains were constructed using the ML method, which showed one major clade of supergroup A and two major clades of supergroup B, which could be easily distinguished from other *Wolbachia* strains (Figure 2) and were congruent with UPGMA analysis (Figure 3). In principle, 12 group A and 5 group B Taiwan *Wolbachia* strains were analyzed with 2 outgroup strains and 5 other *Wolbachia* strains belonging to the groups A and B, respectively (Figure 2 and Figure 3). These comparable *Wolbachia* included *Wolbachia* strains from *Ae. aegypti*, *Ae. albopictus*, and *Culex quinquefasciatus* mosquitoes documented in GenBank (Table 1). Results revealed that all Taiwan *Wolbachia* strains constituted a monophyletic clade genetically affiliated to the *Wolbachia* strains of supergroups A (wAlbA) and B (WalbB), respectively. The discrimination from other *Wolbachia* strains could be easily demonstrated in the same group A or B with a bootstrap value of 99 and 100 in both ML and UPGMA analysis, respectively (Figure 2 and Figure 3). These results demonstrated a lower genetic divergence within the same group of *Wolbachia* detected in *Ae. aegypti* mosquitoes from Taiwan, but a higher genetic divergence from other *Wolbachia* groups documented from various biological and geographical origins.

### 3.4. Molecular Identification of Field-Caught Ae. aegypti Mosquitoes of Taiwan

To further identify the Taiwan strain of *Aedes* mosquitoes, a PCR assay was performed by targeting the mitochondrial CO1 gene of selected *Wolbachia*-infected and uninfected *Aedes* mosquitoes. The CO1 gene sequences of six *Aedes* mosquitoes (four *Wolbachia*-infected and two uninfected) from Kaohsiung of Taiwan were genetically analyzed with sixteen other mosquito specimens belonging to three *Aedes* species (*Ae. aegypti*, *Ae. albopictus* and *Ae. flavopictus*) and four *Culex* species (*Cx. quinquefasciatus*, *Cx. tritaeniorhynchus*, *Cx. fuscanus* and *Cx. gelidus*). The results demonstrated that all Taiwan *Aedes* samples were genetically affiliated to the *Ae. aegypti* group, with a high sequence similarity (99.71–100% similarity), and can be obviously discriminated from other strains of *Aedes* and *Culex* mosquitoes (Figure 4). All these *Ae. aegypti* collected from Taiwan were registered and assigned GenBank numbers (OP889677, OP889681 and OP895029-032).

## 4. Discussion

The present study represents the first description regarding the molecular identification of *Wolbachia* endosymbiont in field-caught *Ae. aegypti* mosquitoes collected from southern Taiwan. In this study, the overall *Wolbachia* infection rate (3.3%) that was detected in *Ae. aegypti* mosquitoes collected from Taiwan was lower than the reported infection rates in previous studies, which were described as 16.8% in Manila, Philippines, 25% in Kuala Lumpur, Malaysia, and 57.4% in New Mexico, USA, respectively [28,29,30]. In addition, another study, investigating *Ae. aegypti* mosquitoes from India, also demonstrated the natural occurrence of *Wolbachia* infection [31]. Our detection of *Wolbachia* infection was performed with an individual mosquito sample. However, the *Wolbachia* infection reported by previous studies was identified by testing samples with pooled mosquitoes. Thus, this may partially explain the higher *Wolbachia* infection reported in previous studies. In any case, the present study reveals the first molecular detection of *Wolbachia* endosymbiont existing in *Ae. aegypti* mosquitoes collected from Kaohsiung of Taiwan, and provides the first convincing sequences (GenBank accession numbers: OP882272-83 and OP896740-4) of *Wolbachia* endosymbionts detected in field-collected *Ae. aegypti* mosquitoes of southern Taiwan.

The existence of natural *Wolbachia* infection in *Ae. aegypti* mosquitoes remains controversial. Although *Ae. albopictus* mosquitoes have been verified with natural *Wolbachia* infections [7,26,27], a recent global survey from various countries described the lack of natural *Wolbachia* infections in *Ae. aegypti* [27]. However, numerous recent studies have contradicted this claim, and have provided evidence of natural *Wolbachia* infections in *Ae. aegypti* mosquitoes [28,29,30,31]. In addition, the persistence of a low presence of *Wolbachia* sequences was also detected in the midgut of *Ae. aegypti* mosquitoes [49,50]. Indeed, results from this investigation indicated a low prevalence (3.3%) of natural *Wolbachia* infections in *Ae. aegypti* mosquitoes collected from the fields of southern Taiwan, and the *Wolbachia* infection in male (5.2%) was higher than in the female (2.0%). The reality of this biological characteristic may vary in *Aedes* mosquitoes distributed in various geographical areas or countries. Thus, the present study clearly demonstrated a low prevalence of natural *Wolbachia* infections in field-caught *Ae. aegypti* mosquitoes collected from southern Taiwan and revealed the possibility of the persistence of *Wolbachia* endosymbiont existing in natural *Ae. aegypti* populations.

The genetic group of the *Wolbachia* strain existing in field-collected *Ae. aegypti* mosquitoes needs to be further identified. Although previous reports described how most of the *Wolbachia* strains discovered in *Ae. aegypti* mosquitoes were identified as the supergroup B *Wolbachia* [29,30,31,32], results from the present observation demonstrated that a single *Wolbachia* strain (0.8% of group B and 1.8% of group A) was detected in the majority of the *Wolbachia*-infected *Ae. aegypti* mosquitoes from Taiwan. Only 0.8% (5/665) were simultaneously infected with supergroups A and B of *Wolbachia* endosymbiont (Table 2). The possible mechanisms regarding the modification rescue property [51] and the association with the bacteriophage WO infection [52] have been described with regard to the presence of supergroup A and co-infection with supergroups A and B in *Ae. aegypti* mosquitoes. In addition, the present study also revealed a higher *Wolbachia* infection in male *Ae. aegypti*, and whether this observation may explain the high possibility of maternal transmission of *Wolbachia* in the natural *Ae. aegypti* mosquito population requires further investigation. Thus, the genetic variation of *Wolbachia* strains in field-collected *Ae. aegypti* mosquitoes distributed in different geographical areas or countries needs to be further classified.

The genetic affiliation of *Wolbachia* strains detected in *Ae. aegypti* mosquitoes can be classified by comparing the sequence similarity of the *wsp* gene of the *Wolbachia* endosymbiont. Indeed, the sequence comparison of the *wsp* gene of *Wolbachia* endosymbiont has been shown to be useful for determining the genetic affiliation of *Wolbachia* strains among various species of arthropod hosts [37,42]. In the present study, the phylogenetic analysis of the *wsp* gene from *Ae. aegypti* mosquitoes of Taiwan displayed a high genetic similarity associated with the supergroups A and B (Figure 2 and Figure 3). The *Wolbachia* strains of group A are mainly affiliated with the wAlbA strain identified from *Aedes albopictus* (GenBank accession no. KY817476), and the *Wolbachia* strains of group B are affiliated with the WalbB strain identified from either *Ae. albopictus* (GenBank accession no. AF020059) or *Aedes aegypti* (GenBank accession no. MF999264). The phylogenetic trees constructed by either the Maximum Likelihood (ML) method or the unweighted pair group with arithmetic mean analysis (UPGMA) strongly support genetic discrimination, recognizing the separation of different supergroups between the *Wolbachia* strains detected in *Ae. aegypti* mosquitoes collected from Taiwan and other supergroups of *Wolbachia* strains from different biological and geographical origins. Accordingly, results from this study reveal that genetic identities of *Wolbachia* endosymbionts detected in field-caught *Ae. aegypti* collected from southern Taiwan were classified as a monophyletic group which was genetically affiliated to the supergroups A and B of *Wolbachia* endosymbionts. Further study should deepen the molecular analysis of various target genes of *Wolbachia* to reveal the reality of *Wolbachia* supergroups.

Due to the detrimental effects of *Wolbachia* endosymbiont on mosquito reproduction and pathogen replication, it is interested to evaluate the possible application of *Wolbachia* infections in natural mosquito populations. Indeed, mosquitoes infected with specific *Wolbachia* strains (wMel and wAlbB) have shown the ability to inhibit/limit a variety of human pathogens in mosquitoes, including dengue, chikungunya, zika and *Plasmodium* [12,13,14,15,16,17,18,19,20,21], and *Wolbachia* endosymbionts can be transmitted vertically from infected females to their offspring. These inherited *Wolbachia* can manipulate the host population through cytoplasmic incompatibility (CI) to regulate the mosquito’s reproduction. In general, when *Wolbachia*-infected males mate with females which are uninfected or harboring a different *Wolbachia* type, early embryo death occurs [16,20]. In addition, *Wolbachia*-induced CI has been used as a proposed strategy for reducing the mosquito population in the field by releasing laboratory-produced *Wolbachia*-infected males [9,11,17]. Indeed, field releases of *Wolbachia*-infected males of *Aedes* mosquitoes have been tested in several countries, including Australia, China, Singapore, the USA and Italy, and have significantly reduced the population densities of wild *Aedes* mosquito in the field [53,54,55,56]. However, there are still local differences between *Ae. aegypti* populations and the variation in persistence of *Wolbachia* infection in the field mosquitoes. Indeed, our study also found a low prevalence of *Wolbachia* infection in natural *Aedes aegypti*. Thus, any efforts or attempts to apply this *Wolbachia*-induced strategy in suppressing wild *Aedes* mosquito populations requires further testing and geographical analysis for promising adequate applications in different areas or countries.

In recent decades, dengue fever infection has been recognized as the major mosquito-borne human infection in Taiwan, and there were significant outbreaks of human infections with dengue fever during the years 2014–2015 that resulted in hundreds of deaths in southern Taiwan [57]. Although there have been no subsequent outbreaks of dengue fever in that region since then, sporadic human infections of domestic and imported human cases have still been reported in the following years. Although *Ae. aegypti* mosquitoes are incriminated as the main vector for the transmission of dengue virus, the mass spraying of insecticides for adult mosquitoes and the reduction of breeding sources for larval mosquitoes are routinely used as the traditional control strategies in Taiwan. It is postulated that transinfection of a suitable *Wolbachia* strain into local *Ae. aegypti* mosquitoes may cause the suppression of the *Ae. aegypti* population in a local environment [9,10,11,53,54,55,56]. However, the ability for laboratory mass-reared *Wolbachia*-infected males to compete with wild males for wild females and the adequate ratio for releasing *Wolbachia*-infected males are critical for the evaluation of the feasibility of this method in the natural environment. Thus, an open field trial is necessary for analyzing the possibility of applying this strategy by releasing *Wolbachia*-infected males to mate with females in the natural environment, and to follow up the suppressive impacts on *Aedes aegypti* populations in Taiwan.

## 5. Conclusions

This study describes the first molecular detection and genetic classification of the *Wolbachia* endosymbionts discovered in field-caught *Ae. aegypti* mosquitoes collected from southern Taiwan. Phylogenetic analysis based on the *wsp* gene of *Ae. aegypti* mosquitoes revealed them to be either singly or superinfected with both groups A and B of *Wolbachia* endosymbionts. In addition, this investigation also describes strong evidence of new findings of group A *Wolbachia* detected in field-collected *Ae. aegypti* mosquitoes. Due to the possible application of *Wolbachia* endosymbionts for the biological control of mosquito populations, the potential role of *Wolbachia* endosymbionts in vector mosquitoes and their microbiome interactions within mosquitoes need to be further identified.

## Figures and Tables

**Figure 1 microorganisms-11-01911-f001:**
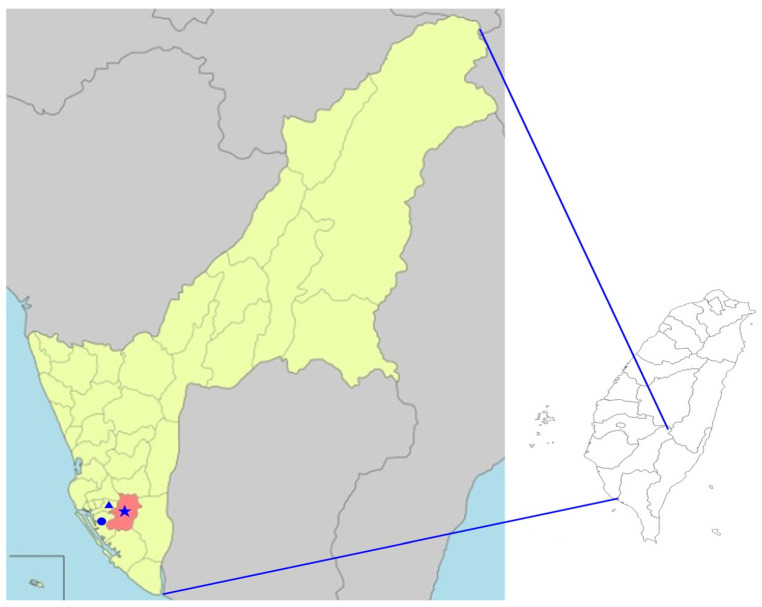
Map of Kaohsiung City of Taiwan showing the mosquito collection sites from 3 districts of Fongshan (★), Cianjhen (●), and Lingya (▲) in Kaohsiung City.

**Figure 2 microorganisms-11-01911-f002:**
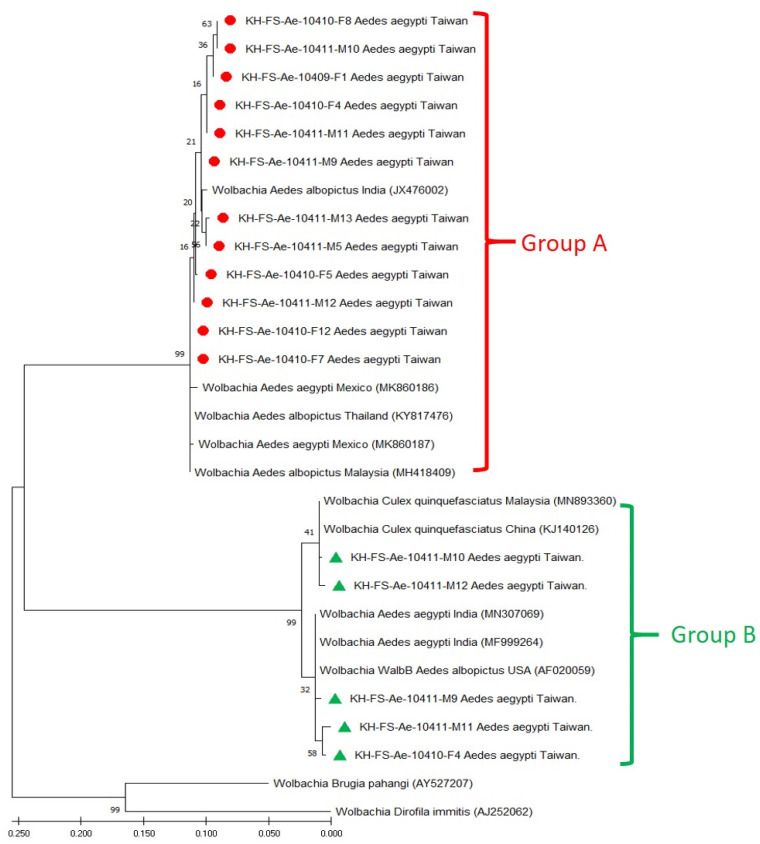
Phylogenetic relationships based on the *Wolbachia* surface protein (*wsp*) gene sequences from 12 specimens of group A (indicated as ●) and 5 specimens of group B (indicated as ▲) of *Aedes aegypti* collected from Taiwan, compared with 12 other specimens belonging to supergroups A and B and outgroup *Wolbachia* documented in GenBank. The tree was constructed and analyzed with the Maximum Likelihood method using 1000 bootstraps replicates. Numbers at the nodes indicate the percentages of reliability of each branch of the tree. Branch length is drawn proportional to the estimated sequence divergence.

**Figure 3 microorganisms-11-01911-f003:**
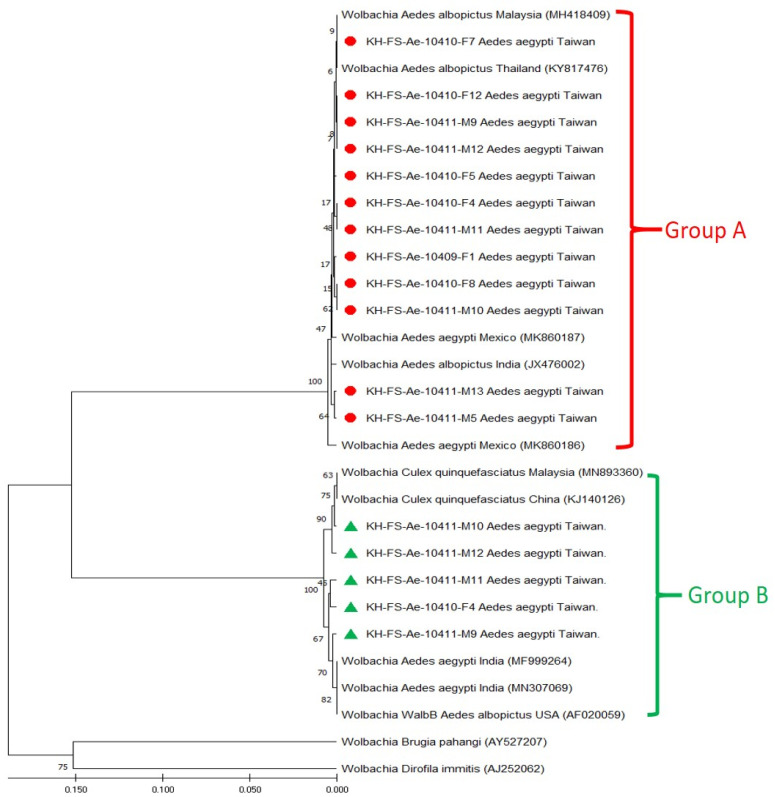
Phylogenetic relationships based on the *Wolbachia* surface protein (*wsp*) gene sequences from 12 specimens of group A (indicated as ●) and 5 specimens of group B (indicated as ▲) of *Aedes aegypti* collected from Taiwan, compared with 12 other specimens belonging to supergroups A and B and outgroup *Wolbachia* documented in GenBank. The tree was constructed and analyzed with the UPGMA method using 1000 bootstrap replicates. Numbers at the nodes indicate the percentages of reliability of each branch of the tree. The branch length was drawn proportional to the estimated sequence divergence.

**Figure 4 microorganisms-11-01911-f004:**
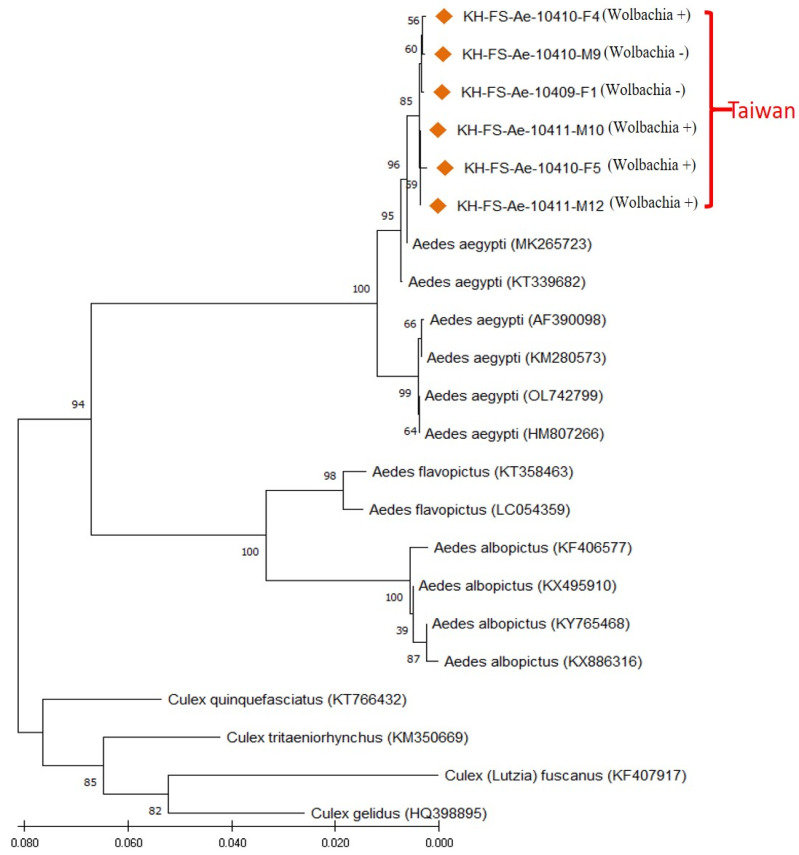
Phylogenetic relationships based on the mosquito CO1 gene sequences from 6 specimens of *Aedes aegypti* collected from Taiwan (indicated with ◆) were compared with 16 specimens of various *Aedes* and *Culex* species documented in GenBank. The tree was constructed and analyzed with the Neighbor-Joining method using 1000 bootstraps replicates. Numbers at the nodes indicate the percentages of reliability of each branch of the tree. The branch length is drawn proportional to the estimated sequence divergence.

**Table 1 microorganisms-11-01911-t001:** *Wolbachia* strains used for phylogenetic analysis in this study.

Genogroup and Strain	Origin of *Wolbachia* Strain		*wsp* Gene
	Biological	Geographic	Accession Number ^a^
Supergroup A			
KH-FS-Ae-10409-F1	*Aedes aegypti*	Kaohsiung, Taiwan	OP882272
KH-FS-Ae-10410-F4	*Aedes aegypti*	Kaohsiung, Taiwan	OP882273
KH-FS-Ae-10410-F5	*Aedes aegypti*	Kaohsiung, Taiwan	OP882274
KH-FS-Ae-10410-F7	*Aedes aegypti*	Kaohsiung, Taiwan	OP882275
KH-FS-Ae-10410-F8	*Aedes aegypti*	Kaohsiung, Taiwan	OP882276
KH-FS-Ae-10410-F12	*Aedes aegypti*	Kaohsiung, Taiwan	OP882277
KH-FS-Ae-10411-M5	*Aedes aegypti*	Kaohsiung, Taiwan	OP882278
KH-FS-Ae-10411-M9	*Aedes aegypti*	Kaohsiung, Taiwan	OP882279
KH-FS-Ae-10411-M10	*Aedes aegypti*	Kaohsiung, Taiwan	OP882280
KH-FS-Ae-10411-M11	*Aedes aegypti*	Kaohsiung, Taiwan	OP882281
KH-FS-Ae-10411-M12	*Aedes aegypti*	Kaohsiung, Taiwan	OP882282
KH-FS-Ae-10411-M13	*Aedes aegypti*	Kaohsiung, Taiwan	OP882283
wol 5/Odisha	*Aedes albopictus*	India	JX476002
AlbA03	*Aedes aegypti*	Mexico	MK860186
AP66-1W	*Aedes albopictus*	Thailand	KY817476
wAlbA04	*Aedes aegypti*	Mexico	MK860187
F8A	*Aedes albopictus*	Malaysia	MH418409
Supergroup B			
KH-FS-Ae-10410-F4	*Aedes aegypti*	Kaohsiung, Taiwan	OP896740
KH-FS-Ae-10411-M9	*Aedes aegypti*	Kaohsiung, Taiwan	OP896741
KH-FS-Ae-10411-M10	*Aedes aegypti*	Kaohsiung, Taiwan	OP896742
KH-FS-Ae-10411-M11	*Aedes aegypti*	Kaohsiung, Taiwan	OP896743
KH-FS-Ae-10411-M12	*Aedes aegypti*	Kaohsiung, Taiwan	OP896744
Kla6	*Culex quinquefasciatus*	Malaysia	MN893360
GD13098	*Culex quinquefasciatus*	China	KJ140126
wAegB	*Aedes aegypti*	India	MN307069
wAegB	*Aedes aegypti*	India	MF999264
WalbB	*Aedes albopictus*	USA	AF020059
Outgroup			
TRS	*Brugia pahangi*	USA	AY527207
unknown	*Dirofilaria immitis*	Italy	AJ252062

^a^ GenBank accession numbers (OP882272-83 and OP896740-4) were submitted by this study.

**Table 2 microorganisms-11-01911-t002:** Detection of *Wolbachia* infection in wild-caught *Aedes aegypti* mosquitoes collected from southern Taiwan with nested-PCR assay targeting the *Wolbachia* surface protein (*wsp*) gene.

Sex of Mosquito	No. Examined	Wolbachia-Group Infection			Total	
		A	B	A&B	No. Infected	% Infection
Female	395	6	1	1	8	2.0
Male	270	6	4	4	14	5.2
Total	665	12	5	5	22	3.3
(%)		(1.8)	(0.8)	(0.8)		

**Table 3 microorganisms-11-01911-t003:** Intra- and inter-group analysis of genetic distance values ^a^ based on the *wsp* gene sequences between the **group A** *Wolbachia* strains of Taiwan and other *Wolbachia* strains belonging to the supergroups A and B and outgroup documented in GenBank.

*Wolbachia* Strains ^b^	1	2	3	4	5	6	7	8	9	10	11	12	13	14	15	16	17
1. *Wolbachia* Thailand (KY817476)	—																
2. *Wolbachia* Malaysia (MH418409)	0.000	—															
3. *Wolbachia* India (JX476002)	0.000	0.000	—														
4. KH-FS-Ae-10410-F12 (Taiwan)	0.000	0.000	0.000	—													
5. KH-FS-Ae-10410-F7 (Taiwan)	0.000	0.000	0.000	0.000	—												
6. KH-FS-Ae-10411-M9 (Taiwan)	0.000	0.000	0.000	0.000	0.000	—											
7. KH-FS-Ae-10410-F8 (Taiwan)	0.000	0.000	0.000	0.000	0.000	0.000	—										
8. KH-FS-Ae-10410-F4 (Taiwan)	0.000	0.000	0.000	0.000	0.000	0.000	0.000	—									
9. KH-FS-Ae-10410-F5 (Taiwan)	0.000	0.000	0.000	0.000	0.000	0.000	0.000	0.000	—								
10. KH-FS-Ae-10411-M11 (Taiwan)	0.000	0.000	0.000	0.000	0.000	0.000	0.000	0.000	0.000	—							
11. KH-FS-Ae-10411-M12 (Taiwan)	0.000	0.000	0.000	0.000	0.000	0.000	0.000	0.000	0.000	0.000	—						
12. KH-FS-Ae-10409-F1 (Taiwan)	0.000	0.000	0.000	0.000	0.000	0.000	0.000	0.000	0.000	0.000	0.000	—					
13. KH-FS-Ae-10411-M10 (Taiwan)	0.000	0.000	0.000	0.000	0.000	0.000	0.000	0.000	0.000	0.000	0.000	0.000	—				
14. KH-FS-Ae-10411-M13 (Taiwan)	0.006	0.000	0.006	0.006	0.006	0.006	0.006	0.006	0.006	0.006	0.006	0.006	0.006	—			
15. KH-FS-Ae-10411-M5 (Taiwan)	0.006	0.006	0.006	0.006	0.006	0.006	0.006	0.006	0.006	0.006	0.006	0.006	0.006	0.000	—		
16. *Wolbachia* India (MF999264)	0.518	0.518	0.518	0.518	0.518	0.518	0.498	0.518	0.518	0.518	0.518	0.518	0.518	0.532	0.532	—	
17. *Wolbachia* B. pahangi (AY527207)	0.788	0.788	0.788	0.788	0.788	0.788	0.793	0.788	0.788	0.788	0.788	0.788	0.788	0.804	0.804	1.093	—

^a^ The pairwise distance calculation was performed using the method of Kimura 2-parameter, as implemented in MEGA X [46]. ^b^ Strains 1–3, 16 and 17 are the documented *Wolbachia* strains of supergroups A and B and outgroup, respectively.

**Table 4 microorganisms-11-01911-t004:** Intra- and inter-group analysis of genetic distance values ^a^ based on the *wsp* gene sequences between the **group B** *Wolbachia* strains of Taiwan and other *Wolbachia* strains belonging to the supergroups B and A and outgroup documented in GenBank.

*Wolbachia* Strains ^b^	1	2	3	4	5	6	7	8	9	10	11	12	13
1. *Wolbachia* WalbB USA (AF020059)	—												
2. *Wolbachia* India (MF999264)	0.000	—											
3. *Wolbachia* India (MN307069)	0.000	0.000	—										
4. KH-FS-Ae-10411-M9 (Taiwan)	0.000	0.000	0.000	—									
5. KH-FS-Ae-10410-F4 (Taiwan)	0.000	0.000	0.000	0.000	—								
6. KH-FS-Ae-10411-M11 (Taiwan)	0.000	0.000	0.000	0.000	0.000	—							
7. KH-FS-Ae-10411-M10 (Taiwan)	0.008	0.008	0.008	0.008	0.008	0.008	—						
8. KH-FS-Ae-10411′-M12 (Taiwan)	0.008	0.008	0.008	0.008	0.008	0.008	0.000	—					
9. *Wolbachia* China (KJ140126)	0.018	0.018	0.018	0.008	0.008	0.008	0.000	0.000	—				
10. *Wolbachia* Malaysia (MN893360)	0.018	0.018	0.018	0.008	0.008	0.008	0.000	0.000	0.000	—			
11. *Wolbachia* Thailand (KY817476)	0.417	0.417	0.417	0.302	0.302	0.302	0.286	0.286	0.415	0.415	—		
12. *Wolbachia* B. pahangi (AY527207)	0.768	0.768	0.768	0.527	0.527	0.527	0.515	0.515	0.677	0.677	0.559	—	
13. *Wolbachia* D. immitis (AJ252062)	0.602	0.602	0.602	0.482	0.482	0.482	0.480	0.480	0.574	0.574	0.681	0.369	—

^a^ The pairwise distance calculation was performed using the method of Kimura 2-parameter, as implemented in MEGA X [46]. ^b^ Strains 1–3 and 9, 10 are the documented *Wolbachia* strains of supergroup B; strains 11 and 12, 13 are documented supergroup A and the outgroup, respectively.

## Data Availability

All data described in this paper are available after publication.

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
