# Peer review of "First Detection and Genetic Identification of Wolbachia Endosymbiont in Field-Caught Aedes aegypti (Diptera: Culicidae) Mosquitoes Collected from Southern Taiwan"

_microorganisms, 2023, doi:10.3390/microorganisms11081911_

Round 1
Reviewer 1 Report
The authors present for the first time the possible presence of the intracellular bacterium Wolbachia in natural populations of Aedes aegypti in Taiwan. The investigation of the Wolbachia bacterium arouses much interest in the scientific community due to its ability to induce cytoplasmic incompatibility in various insects, including mosquitoes, and therefore its strong impact on the biological control of disease transmitted by vectors insects and insect pests. However, as pointed out by the authors themselves, there is much controversy on the subject. In literature, there are now numerous works on discovering Wolbachia by molecular investigations on different kinds of mosquitoes. Considering the low percentage compared to the natural population analyzed, these sequences could be simple insertions. Also, in this work, there are no microscopy images that support the molecular data. It is also now known that analyzing a single gene is not enough.
Considering the small number of samples that tested positive for the Wolbachia wsp gene, I would suggest the authors amplify and analyze at least the MLST genes of the bacterium (Baldo et al., 2006) for a better analysis of the phylogeny.
Minor comments:
In M&M section: enter the coordinates and a map of the sampling sites;
insert a table of the oligonucleotides used;
line 148: change biologiical with biological
Author Response
General comments: The reviewer enquires about no microscopy images to support the molecular data and whether the phylogenetic analysis based on a single wsp gene is enough.
Reply: We are working on the molecular detection and identification of Wolbachia detected in the natural Aedes aegypti mosquitoes, but not a cell culture isolation. In addition, wsp gene is dominant and species-specific gene for Wolbachia, and it can be used for differential diagnosis from other Rickettsia organisms. Current phylogenetic analysis of Wolbachia groups based on the wsp gene is recognized as a reliable method for molecular identification (Werren et al., 1995, Proc. R. Soc. Lond. Biol. Sci., 261, pp.55-63; Zhou et al., 1998, Proc. R. Soc. Lond. Biol. Sci., 265, pp.509-515; Rueng-Areerate et al., 2003, J. Med. Entomol., 40, pp.1-5).
Minor comments:
In M&M section: enter the coordinates and a map of the sampling sites.
Reply: As requested, we have added one Taiwan map indicates the collection sites in Kaohsiung City (Figure 1; Page 3, Lines 90-92).
Insert a table of the oligonucleotides used.
Reply: We have added the primer’s sequences in the adequate position of M&M section showing the oligonucleotides used in this study (Page 3, Lines 107-112).
Line 148: change biologiical with biological.
Reply: We appreciate the reviewer’s correction and have done it (Line 154)..
Reviewer 2 Report
This manuscript details the experiments conducted to examine the presence of naturally occurring Wolbachia infection in Aedes aegypti collected in a few locations in Southern Taiwan. The experiments are generally well conducted, and the study is appropriately written and referenced. A few items and comments to improve the manuscript are noted below. None are major issues, primarily just missing information and suggestions to improve the utility of the study. Additional copyediting for spelling and standard English usage would improve the readability.
P1L42-44: Suggest the reference call outs need to be increased to 12-18 to include the reference to Plasmodium. Also, it is necessary to replace Aedes with Aedes aegypti to be more specific to the point being made. wAlbB is natively found in Aedes albopictus thus no need to be introduced.
L54: Reference numbering seems to be slightly off. Should reference 33 be included here or should this stop at 32? Reference 33 is about deer ticks. Please review all reference callouts.
L87-88: Good to see that species ID of the mosquito was confirmed by CO1. What was the cutoff percentage for matching/non-matching? Where were the ground truth sequences for comparison found, GenBank or BOLDv4?
L144: Please provide information about the thresholds used regarding minimum sequence quality. What was the minimum PHRED score used before the sample was included in the analysis? Was the sequence a consensus of both a clean forward and reverse read or just a single direction read? If single, how did you deal with the low-quality sequence in the initial positions in the data?
P12L64-65: This sentence seems to be an orphan. What is missing?
P12L81-83: While the infection in males was nominally higher than females, it is overly speculative to say that there may be gender differences based on such small numbers. Suggest it be removed unless you can support similar using references. This includes L98-100.
L122-141: This paragraph provides very good information about Wolbachia based control strategies. Much of it is more appropriate for the intro. This paragraph should directly address the big question. What is the implication of finding natural Wolbachia infection on these control strategies (whether this or the paragraph below)? Will it improve or is it likely to hurt these efforts? What have the other studies that have detected natural infection in Aedes aegypti suggested to this issue? If mating with a naturally infected female is generally non-productive, how will this change the dynamics of driving an artificial infection into the natural population? Much to consider and it would be good to see these potential questions discussed.
See comments to Authors.
Author Response
P1L42-44: Suggest the reference need to be increased to 12-18 to include the reference to Plasmodium. Also, it is necessary to replace Aedes with Aedes aegypti to be more specific to the point. wAlbB in Aedes albopictus is no need to be introduced.
Reply: We appreciate the reviewer’s comments and have done it (Page 1, Lines 42-44).
L54: Should reference 33 be included here or should stop at 32?
Reply: We appreciate the reviewer’s correction and have done it (Page 2, Line 54).
L87-88: What was the cutoff percentage for matching/non-matching? Where were the comparison sequences found, GenBank or BOLDv4?
Reply: The matching principle is based on the primer binding sequence after alignment of all the comparison sequences. All the comparison sequences are available from the database documented in GenBank (Page 2, Lines 87-88).
L144: Please provide information about the thresholds used regarding minimum sequence quality. What was the minimum PHRED score used before the sample was included in the analysis? Was the sequence a consensus of both a clean forward and reverse read or just a single direction read? How did you deal with the low-quality sequence in the initial positions in the data?
Reply: All our samples for sequencing were submitted to the professional technical company (Mission Biotech Co., Ltd., Taiwan) for DNA sequencing (Page 4, Lines 145-146). The thresholds used for minimum sequence quality depend on the company’s optimal condition. The minimum PHRED score of >40 was used. The sequence was read by a single direction of forward primer and the low-quality sequence will give up from the data.
P12L64-65: This sentence seems to be an orphan. What is missing?
Reply: We appreciate the reviewer’s comments and have modified it (Page 13, Lines 65-66).
P12L81-83: It is overly speculative to say that there may be gender differences based on such small numbers. Suggest it be removed. This includes L98-100.
Reply: We appreciate the reviewer’s comments and have done it (Page 13, Lines 82-84; 99-101).
L122-141: What is the implication of finding natural Wolbachia infection on these control strategies (whether this or the paragraph below)? Will it improve or is it likely to hurt these infection in Aedes aegypti suggested to this issue?
Reply: We appreciate the reviewer’s comments and have modified the sentences (Page 14, Lines 139-140; 158-160).
Round 2
Reviewer 1 Report
I thank the authors for accepting my suggestions. However, they have not responded to the significant criticality in the manuscript. I know wsp is Wolbachia-specific and dominant. However, considering the possibility of portions of genes that can be inserted into the mosquito genome, it is now standard practice to identify at least the 5 genes of the Wolbachia MLST. As far as FISH experiments are concerned, there is no need to work on cell cultures. FISH is usually performed on dissected reproductive organs. However, considering the very low prevalence of Wolbachia found, I think it is sufficient to deepen the molecular analysis on the few samples that tested positive. I reiterate that it is essential to try to identify the other 5 genes.
Author Response
The reviewer suggests that we should do Wolbachia MLST. As far as FISH experiments are concerned, there is no need to work on cell cultures. FISH is usually performed on dissected reproductive organs.
Reply: We appreciate the reviewer’s comments. As we know, the various target genes have different sensitivity for detection of Wolbachia within mosquitoes and there are no consistent results for the detection of Wolbachia by different target genes. However, the detection based on wsp gene is most reliable gene used for Wolbachia sequence comparison and the wsp gene sequences (either biological or geographical origins) are most available from GenBank.
The FISH experiments need to dissect the reproductive organs for fluorescent In situ hybridization, but our purpose is not focused on the tissue tropism of Wolbachia organism. In addition, the FISH experiments remain need sequence data to verify the Wolbachia organism. To compry with the reviewer’s concern, we have added one sentence in the Discussion section regarding the further study should deepen the molecular analysis on various target genes to reveal the reality of Wolbachia supergroups (Page 14, Lines 122-123).